# The Multi-Orbits Skew Spectrum: Boosting Permutation-Invariant Data Representations

**Armando Bellante**
Dipartimento di Elettronica, Informazione e Bioingegneria, Politecnico di Milano
armando.bellante@polimi.it

**Martin Plávala**
Naturwissenschaftlich-Technische Fakultät, Universität Siegen
martin.plavala@uni-siegen.de

**Alessandro Luongo**
Centre for Quantum Technologies, National University of Singapore
Inveriant Pte. Ltd.
ale@nus.edu.sg

## Abstract

We generalize the concept of skew spectrum of a graph, a group-theoretical permutation-invariant feature mapping. The skew spectrum considers adjacency matrices as functions over $\mathbb{S}_n$ and leverages Fourier transform and group-theoretical tools to extract features that are invariant under the group action. The main shortcoming of the previous result is that the skew spectrum only works for unlabeled graphs. The reason is that these graphs can be represented using matrices whose main diagonal contains zeros, meaning that there is only one set of elements that can permute among themselves (i.e., one orbit). However, the representations of more complex graphs (e.g., labeled graphs, multigraphs, or hypergraphs) have different sets of elements that can consistently permute on different orbits. In this work, we generalize the skew spectrum to the multiple orbits case. Our multi-orbits skew spectrum produces features invariant to such permutations and possibly informative of non-consistent ones. We show this method can improve the performances of models that learn on graphs by providing comparisons with single orbit representations and eigenvalues. Moreover, the theory is general enough to handle invariance under the action of any finite group on multiple orbits and has applications beyond the graph domain.

## 1   Introduction

In the past decade, machine learning on datasets representing data as tensors became an active area of research. This trend is fueled by important applications in anomaly detection, medical imaging, genomics, and many others [1–3]. We generalize the skew spectrum, a graph invariant proposed in [4], to a new tensor invariant. In the context of graphs, for which we frame this manuscript, our generalization extends the applicability of this feature extraction method to more complex data structures, such as labelled graphs, multigraphs, and hypergraphs.

The skew spectrum of a graph is a permutation invariant mapping from the adjacency matrix $A \in \mathbb{R}^{n \times n}$ of a weighted, directed, *unlabeled* graph $\mathcal{G}$ to a new feature space. A graph is unweighted when the entries of $A$ are just elements of $\{0, 1\}$, undirected when $A^T = A$, and unlabeled when $A_{ii} = 0$ for $i \in [n]$. This mapping interprets the graph as a function on the symmetric group $f : \mathbb{S}_n \mapsto \mathbb{R}$, where $n$ is the number of nodes in the graph. The function $f$ is defined as $f(\sigma) = A_{\sigma(n), \sigma(n-1)}$, for

A. Bellante et al., The Multi-Orbits Skew Spectrum: Boosting Permutation-Invariant Data Representations (Extended Abstract). Presented at the First Learning on Graphs Conference (LoG 2022), Virtual Event, December 9–12, 2022.

$\sigma \in \mathbb{S}_n$, here $\sigma \in \mathbb{S}_n$ is permutation of the set $[n]$ and $\sigma(n)$ is the image of $n$ under the permutation $\sigma$. For more precise definitions on graph theory we refer the reader to Appendix A. Leveraging techniques from non-commutative harmonic analysis it is possible to see that the skew spectrum of a function $f$ is related to the Fourier transform of the triple correlation of $f$ [5]. An entry of the skew spectrum is a matrix, denoted as $\mathcal{S}_f(\sigma, \rho)$, which is a function of a permutation $\sigma \in \mathbb{S}_n$, and an irreducible representation $\rho$. We recall that a representation of $\mathbb{S}_n$ is a map $\rho$ from the group $\mathbb{S}_n$ to a subgroup of the orthogonal group on a real, finite-dimensional vector space. The representation $\rho$ is irreducible if it cannot be decomposed into direct sum of other representations. The intuition here is that $\mathcal{S}_f(\sigma, \rho)$ is a polynomial of third order in the adjacency matrix such that this polynomial is invariant with respect to joint permutations of rows and columns of the adjacency matrix. While the skew spectrum might be a *complete* invariant in some cases [6], this is not true for many many applications of the skew spectrum to permutation-invariant representations (i.e., two non-isomorphic graphs could be mapped to the same feature vector).

The reduced skew spectrum [4, Definition 2] is a lightweight version of the skew spectrum, which is defined by reducing the size of the matrices of $\mathcal{S}_f(\sigma, \rho)$. The motivation for using the reduced skew spectrum is threefold: While the computation of the skew spectrum has a complexity of $O(n^6)$, the reduced skew spectrum has a computational complexity of $O(n^3)$ only (here $n$ is the number of the nodes of the graph); the output size of the reduced skew spectrum is independent of $n$ and skew spectrum contains many entries which are trivially zero, the reduced skew spectrum eliminates almost all such entries. The (reduced) skew spectrum can only be applied to datasets with a single orbit; this limitation is particularly important when dealing with tensor datasets. In this work, we extend the skew spectrum to the multi-orbits setting. We show how we can inherit good computational properties of the reduced skew spectrum also for the multiple orbit setting. While the main contribution of this work is theoretical, we corroborate our analysis with some prototypical experiments on real and synthetic datasets. We conclude that the multiple-orbits skew spectrum can enhance the representation of datasets where keeping a consistency between permutation on different orbits is important.

## 2 The multi-orbits skew spectrum

The main idea in generalizing the skew spectrum to multiple orbits is two-fold: we replace the function $f : \mathbb{S}_n \mapsto \mathbb{R}$ by a vector-valued function $f : \mathbb{S}_n \mapsto \mathbb{R}^k$, where $k$ is the number of orbits, and we replace products of functions by tensor products. For simplicity, we will describe the computation for the case of two orbits for a function generated by the adjacency matrix $A \in \mathbb{R}^{n \times n}$ of a graph $\mathcal{G}$. Note, however, that all of our formulas are valid for any finite number of orbits. The interested reader can refer to Appendix A for the representing more complex graphs as functions.

A labeled graph is one for which the adjacency matrix has some non-zero entries on the diagonal. For a given labeled graph $\mathcal{G}$, the adjacency matrix $A$ is unique only up to arbitrary permutations of the same indexes of both rows and columns. If we can obtain the adjacency matrix of a graph $\mathcal{G}$ by applying a permutation $\sigma \in \mathbb{S}_n$ to the indexes of the adjacency matrix of another graph $\tilde{\mathcal{G}}$, then we say the graphs are isomorphic. It is fairly easy to see that the adjacency matrix of a labeled graph has two orbits: the main diagonal, and the off-diagonal. For $\sigma \in \mathbb{S}_n$, let $f : \mathbb{S}_n \mapsto \mathbb{R}^2$ be defined as

$$f(\sigma) = \begin{pmatrix} A_{\sigma(n),\sigma(n)} \\ A_{\sigma(n),\sigma(n-1)} \end{pmatrix}. \tag{1}$$

We define an entry of the multi-orbits skew spectrum as

$$\mathcal{S}_f(\sigma, \rho) = \frac{1}{(n!)^2} \sum_{\tilde{\sigma}_1 \in \mathbb{S}_n} \sum_{\tilde{\sigma}_2 \in \mathbb{S}_n} f(\tilde{\sigma}_1) \otimes f(\tilde{\sigma}_1 \sigma) \otimes f(\tilde{\sigma}_2) \otimes \left( \rho(\tilde{\sigma}_1)^\dagger \rho(\tilde{\sigma}_2) \right), \tag{2}$$

where $\dagger$ denotes the usual complex conjugate transpose. From this formula, one can notice that the skew spectrum entries are invariant to permutations of the indices of the adjacency matrix, see Appendix B for a formal proof and some intuition on the invariance.

Naively computing one skew spectrum entry from Eq. 2 would require $O((n!)^2)$ steps, which soon becomes computationally infeasible even for small graphs. However, we can significantly speed up the calculation using insights from group theory. Up to improvements of constant factors, our computational speedups coincide with the ones presented in Kondor and Borgwardt [4] and so the computational cost is $O(n^6)$ (we are considering $k = 2$ as a constant). The key insight on the

computational optimization that we adapt from the single-orbit case is the following. Denoting $\hat{f}$ the Fourier transform of $f$ and writing

$$\hat{f}(\rho) = \frac{1}{n!} \sum_{\tilde{\sigma} \in \mathbb{S}_n} f(\tilde{\sigma}) \otimes \rho(\tilde{\sigma}) \quad \text{and} \quad \hat{r}_f(\sigma, \rho) = \frac{1}{n!} \sum_{\tilde{\sigma} \in \mathbb{S}_n} f(\tilde{\sigma}) \otimes f(\tilde{\sigma}\sigma) \otimes \rho(\tilde{\sigma}) \qquad (3)$$

we get that Eq. 2 can be rewritten as $\mathcal{S}_f(\sigma, \rho) = \hat{r}_f^\dagger(\sigma, \rho) \odot \hat{f}(\rho)$, where $\odot$ denotes tensor product of functions and matrix product of representations. This significantly speeds up the calculation since the calculation of $\hat{f}$ can be reduced to sum only over $n(n-1)$ elements (as a property of $f$) and the calculation of $\hat{r}_f$ can be reduced in a similar way, via Clausen-FFT type of arguments [5, 7].

At the same time, Kondor and Borgwardt [4] show that the skew spectrum needs to be computed for only 7 group entries and 4 irreducible representations, fixing the overall computation of the skew spectrum to $O(n^6)$. It is possible to prove that we actually need only 6 group elements. Indeed, 2 of the 7 are inverses of each other and computing the skew spectra for both of them does not introduce extra information. Moreover, we have empirical evidence that one of the 4 irreducible representation only produces 0 matrices for undirected graphs and can therefore be discarded in such case.

While the computational complexity is polynomial in the size of the graph, there are still two problems: $O(n^6)$ starts to be infeasible for medium-sized graphs and the number of computed features scales with the number of nodes. Both of these problems are solved by using the reduced skew spectrum [4]. The main idea is that the Fourier transform $\hat{f}$ consists of matrices that have variable number of rows but fixed number of columns, so the variable size of skew spectrum comes only from $\hat{r}_f$. We thus compute the reduced skew spectrum by limiting the number of rows of $\hat{r}_f$, producing matrices of fixed size for all $n$, with a computational complexity of $O(n^3)$. In the general case, with $k$ orbits, the computational complexity of the reduced skew spectrum becomes $O(k^2 n^3 + k^3 n^2)$. For a symmetric adjacency matrix of a graph with $n$ nodes, the single-orbit reduced skew spectrum consists of 36 numbers, while the two-orbits reduced skew spectrum consists of 288 numbers. In general, for $k$ orbits, the reduced multi-orbits skew spectrum consists of $36k^3$ numbers.

**Generalization to multigraphs and hypergraphs.** We present the generalization of our method to multigraphs. Different types of edges can be seen as several adjacency matrices and we can treat those as additional orbits. Consider a multigraph with two types of edges. Let $A$ be the adjacency matrix corresponding to the first type of edges and let $B$ be adjacency matrix corresponding to the second type of edges (note that the diagonals of both $A$ and $B$ coincide since the labels of the nodes are always the same). We can then construct the function $f_{AB}(\sigma) = (A_{\sigma(n),\sigma(n)}, A_{\sigma(n),\sigma(n-1)}, B_{\sigma(n),\sigma(n-1)})^\intercal$ and use the skew spectrum (or reduced skew spectrum) of $f_{AB}$.

We can always represent a hypergraph as a multigraph where each edge type is an edge, but we also propose an alternative encoding. Consider first hyperedges that connect exactly 3 nodes. We can then represent these hyperedges via $n \times n \times n$ adjacency tensor $A$ such that $A_{abc}$, $a \neq b \neq c \neq a$, is the weight of the hyperedge connecting the nodes labeled by $a$, $b$, and $c$. We can then use similar methods to construct the corresponding function $f$ and compute its skew spectrum. We can use similar approach for hypergraphs with hyperedges connecting at most $k$ nodes by considering adjacency tensors $A_{a_1 \ldots a_k}$. With this alternative representation, and still considering the amount of orbits as a constant, the computational complexity of the skew spectrum and the reduced skew spectrum will increase. This is because for a standard graph the computational speedup heavily relies on the fact that the function $f$ is $\mathbb{S}_{n-2}$ symmetric, but, for example, for hypergraph with hyperedges connecting at most 3 nodes the resulting function will be only $\mathbb{S}_{n-3}$ symmetric. Thus, for example, one would need to sum at least $n(n-1)(n-2)$ elements in order to compute $\hat{f}(\rho)$.

## 3 Numerical Experiments

We implemented the code for both the single-orbit skew spectrum and the multi-orbits skew spectrum. In this section, we report some prototypical experiments using their reduced versions on labeled graphs. The experiments aim to show the enhanced representation power of the multi-orbits skew spectrum against its predecessor. Comparison with other state-of-the-art graph invariants is out of the scope of this manuscript and is left for future work. We discuss some extra experiments in Appendix C.

## 3.1 Graph classification on a synthetic dataset

The dataset contains undirected, unweighted, and labeled graphs. The node labels assume all the values between $0$ and $n - 1$, with $n$ the number of nodes. The labels are encoded along the main diagonal of the graphs' adjacency matrices. The dataset consists of four families of graphs, namely 15A, 15B, 6A, and 6B (Fig. 1 illustrates one graph representative per family). Each family contains 1000 isomorphic graphs, meaning that, inside a family, all the graphs are equal up to permutations that are simultaneously edge-preserving and label-preserving. The graphs in 15A and 15B are equivalent up to edge-preserving permutations but are not isomorphic if we also consider the labels permutations, and so are the graphs in 6A and 6B. The task is to classify the four families of graphs correctly.

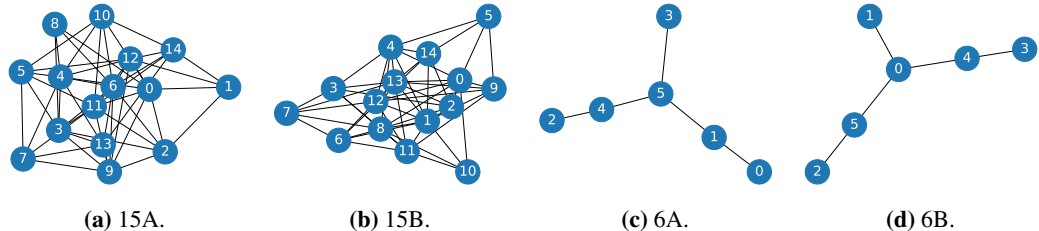

**(a)** 15A.  **(b)** 15B.  **(c)** 6A.  **(d)** 6B.

**Figure 1:** Synthetic dataset's graph families representatives.

We have processed this dataset twice, producing two different representations. The first one consists of a concatenation of the outputs of the single-orbit skew spectrum computed separately on the off-diagonal and on the main diagonal elements of the adjacency matrix (i.e., each skew spectrum uses one row of Eq. 1 as a function). The second one consists of the output of the two-orbits skew spectrum (computed using Eq. 1). We trained a random forest (60 estimators, no max depth) on the two representations, holding out a balanced $20\%$ of the dataset for testing purposes. The classifier achieves an accuracy of $0.5$ on the single-orbit representation and of $1$ on the 2-orbits one, meeting our expectations. This is because the concatenation of single-orbit skew spectra cannot distinguish the couples 15A-15B and 6A-6B, whose labels and edges are linked by different permutations.

## 3.2 Atomization energy regression on QM7

The dataset is composed of a list of $23 \times 23$ matrices representing the Coulomb matrices of 7165 molecules composed of up to 23 atoms, from which up to 7 are considered heavy atoms [8, 9]. The Coulomb matrix $C \in \mathbb{R}^{23 \times 23}$ is defined as $C_{ii} = \frac{1}{2} Z_i^{2.4}$ and $C_{ij} = \frac{Z_i Z_j}{|R_i - R_j|}$, where $Z_i$ is the nuclear charge of the $i$-th atom of the molecule, and $R_i$ is its position. The learning task associated with this dataset is to predict the atomization energies of the molecules (kcal/mol), which are reals in the range $[-2000, -800]$ renormalized in $[-1, 1]$. We compute the reduced single-orbit skew spectrum on the off-diagonal elements and the multiple-orbits skew spectrum and use them for regression, holding out $20\%$ of the dataset for the test set. Table 1 summarizes the results.

**Table 1:** Regression on qm7 with different features. We tested the following machine learning models: Extreme Gradient Boosting (Xgboost), Gradient Boosting Regressor (GBR), Elastic Net (EN), Linear Regression (Linear) using the default parameters of sk-learn[10]. Linear regression cannot fit the dataset for the eigenvectors of the Coulomb matrix. The error is measured as Mean Absolute Error.

| Representation | Xgboost | GBR | EN | Linear |
|---|---|---|---|---|
| Single-orbit | 29.15 | 36.55 | 114.68 | 61.15 |
| 2-orbits | **18.28** | **27.12** | 58.60 | 49.45 |
| $C$'s eigs | 38.04 | 37.92 | 47.83 | - |
| Laplacian's eigs | 23.52 | 26.93 | **47.62** | **47.80** |

# 4 Conclusions

This work presents a generalization of the skew spectrum to multiple orbits. Thanks to our implementation, we can test the performances of the multi-orbits skew spectrum on classification and regression tasks. The performances obtained in these prototypical experiments highlight the limitations of the single-orbit skew spectrum and the advantages of the proposed solution. For the classification experiment, we have a clear separation between the performances of a simple learner on 1-orbit and 2-orbits skew spectra of labeled graphs. For regression, the 2-orbits skew spectrum can improve the mean absolute error compared to the single orbit case of all the machine learning models studied. We leave for future work the study of problems on tensor datasets where the multi-orbits skew spectrum can outperform state-of-the-art machine learning models.

# Acknowledgements

We would like to thank Ramakrishna Kakarala for providing us with a soft copy of his thesis. We thank the Quantum Open Source Foundation, through which we first met. Furthermore, we thank Petar Veličković and Vijay Prakash Dwivedi for the kind feedback and useful discussions. AB would like to thank professors Stefano Zanero, Ferruccio Resta and Donatella Sciuto for their support, and Patrick Rebentrost for hosting him at CQT at the time of writing. MP acknowledges support from the Deutsche Forschungsgemeinschaft (DFG, German Research Foundation, project numbers 447948357 and 440958198), the Sino-German Center for Research Promotion (Project M-0294), the ERC (Consolidator Grant 683107/TempoQ), the German Ministry of Education and Research (Project QuKuK, BMBF Grant No. 16KIS1618K) and from the Alexander von Humboldt Foundation. Research at CQT is funded by the National Research Foundation, the Prime Minister's Office, and the Ministry of Education, Singapore under the Research Centres of Excellence programme's research grant R-710-000-012-135. We also acknowledge funding from the Quantum Engineering Programme (QEP 2.0) under grant NRF2021-QEP2-02-P05.

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

## A    Representing graph data using functions

In this section we describe some cathegories of graphs and how we can represent them using functions of the form $f : \mathbb{S}_n \mapsto \mathbb{R}^k$. Rather than being an exhaustive list, this section should give the reader an idea of how to construct functions for more complex data structures.

The simplest class of graphs that we can represent are directed, weighted graphs with $n$ nodes. Such graphs can be represented using an adjacency matrix $A \in \mathbb{R}^{n \times n}$ with entries equal to 0 on the main diagonal. In this case, we can build a function $f : \mathbb{S}_n \mapsto \mathbb{R}$, as

$$f(\sigma) = A_{\sigma(n),\sigma(n-1)}, \tag{4}$$

where $\sigma(i)$ denotes the image of $n$ under the permutation $\sigma$. The function takes a permutation as input and outputs a matrix element, using the image of $n$ and $n-1$ under this permutation (it only iterates over the non-diagonal elements of $A$). This is the function of the original skew spectrum. Plugging this function in our multi-orbits skew spectrum formulation will give the same result as the single-orbit skew spectrum.

Generalizing to some more complex graph structures, we consider graphs whose nodes can be associated with one attribute.

**Definition 1** (Labeled graph). *Labeled graphs are graphs whose nodes have one attribute that can be encoded as a real number.*

We can represent such graphs using the same adjacency matrix $A \in \mathbb{R}^{n \times n}$, and we could encode the attribute of the node $i$ in the diagonal entry $A_{ii}$. In this case, we want to be sure that a permutation will consistently move both the edges and the node labels. Those live in two different orbits, that need to permute accordingly. In this case, we can construct the graph function $f : \mathbb{S}_n \mapsto \mathbb{R}^2$ as

$$f(\sigma) = \begin{pmatrix} A_{\sigma(n),\sigma(n-1)} \\ A_{\sigma(n),\sigma(n)} \end{pmatrix}. \tag{5}$$

Here the order of the elements in the output vector in $\mathbb{R}^2$ does not matter, as long as the choice is consistent during the computation. Given a permutation $\sigma$, this function returns a vector containing an element on the off-diagonal and an element on the diagonal of $A$.

A more general class of graphs are graphs with node attributes.

**Definition 2** (Graph with node attributes). *A graph with node attributes is a graph whose nodes are associated to a vector of attributes, represented by real numbers.*

Say that each node can contain at most $k'$ distinct attributes. Then, we can represent the graph using an adjacency matrix $A \in \mathbb{R}^{n \times n}$ with zeroes on the main diagonal and a set of $k'$ vectors $x_i \in \mathbb{R}^n$ for $i \in \{0, \dots, n-1\}$. Each vector $x_i$ represents an attribute and contains $n$ entries, one per node. In this case, off-diagonal elements of $A$ can permute among themselves, entries in the same $x_i$ can permute among themselves, but they have to do so consistently, forming $k = k' + 1$ orbits. We represent the $j^{\text{th}}$ entry of $x_i$ as $x_{i,j}$ and it represents the value of the attribute $i$ for the node $j$. We can construct the relative graph function $f : \mathbb{S}_n \mapsto \mathbb{R}^{k'+1}$ as

$$f(\sigma) = \begin{pmatrix} A_{\sigma(n),\sigma(n-1)} \\ x_{1,\sigma(n)} \\ x_{2,\sigma(n)} \\ \dots \\ x_{k',\sigma(n)} \end{pmatrix}. \tag{6}$$

Even in this case, the matrix outputs one element per orbit.

Going on with more complex data structures, we can show how to represent multigraphs.

**Definition 3** (Multigraph). *Multigraphs are graphs where an edge can connect the multiple nodes.*

Say that we have $k$ different layers of edges, each layer having its own meaning. Then, we can represent the graph using a tensor adjacency matrix $A \in \mathbb{R}^{k \times n \times n}$, where $A_{k,i,j}$ represents the value of the edge in layer $k$ between nodes $i$ and $j$. The $k$ layers form $k$ distinct orbits where permutations need to occur consistently. We can construct the graph function $f : \mathbb{S}_n \mapsto \mathbb{R}^k$ as

$$f(\sigma) = \begin{pmatrix} A_{0,\sigma(n),\sigma(n-1)} \\ A_{1,\sigma(n),\sigma(n-1)} \\ \dots \\ A_{k-1,\sigma(n),\sigma(n-1)} \end{pmatrix}. \tag{7}$$

Whenever we wanted to consider multigraphs with node attributes, we could increment the number of orbits and build bigger functions.

About hypegraphs, we refer the reader to the intuition in the main text. We recall that the computation easily starts to become unpractical for hypergraphs having edges connecting more than a handful of nodes (say 6). This is because the formulation for hypergraphs introduces a dependency on $m$ in the runtime (rather than $k$), which is provably at least factorial $m!$.

## B   Invariance of the multi-orbits skew spectrum

**Theorem 4.** *Let $f_1(\sigma) : \mathbb{S}_n \mapsto \mathbb{R}^k$ be a function that maps an element of the permutation group to a vector of real numbers. Let $f_2(\sigma) : \mathbb{S}_n \mapsto \mathbb{R}^k$ be a function that is equivalent to $f_1$ up to input permutation, such that $f_2(\sigma) = f_1(\sigma'\sigma)$ for a fixed $\sigma' \in \mathbb{S}_n$. Then, the multi-orbits skew spectra of $f_2$ and $f_1$ are equal.*

*Proof.* Consider a function $f_1(\sigma) : \mathbb{S}_n \mapsto \mathbb{R}^k$. Now consider a second function $f_2(\sigma) = f_1(\sigma'\sigma)$, which is equivalent to $f_1$ up to a permutation $\sigma' \in \mathbb{S}_n$ of its input. We can show that *each single entry* of the skew spectrum for $f_2$ is equal to the entry for $f_1$, using the fact that $\rho^\dagger(g)\rho(g) = \mathbb{I}$ and that $\sum_{g \in \mathbb{G}} f(g'g) = \sum_{\hat{g} \in \mathbb{G}} f(\hat{g})$ for any group $\mathbb{G}$, any fixed element $g' \in \mathbb{G}$, and any function $f$ defined over the group. We have

$$
\begin{aligned}
\mathcal{S}_{f_2}(\sigma, \rho) &= \frac{1}{(n!)^2} \sum_{\tilde{\sigma}_1 \in \mathbb{S}_n} \sum_{\tilde{\sigma}_2 \in \mathbb{S}_n} f_2(\tilde{\sigma}_1) \otimes f_2(\tilde{\sigma}_1\sigma) \otimes f_2(\tilde{\sigma}_2) \otimes \big(\rho(\tilde{\sigma}_1)^\dagger \rho(\tilde{\sigma}_2)\big) \\
&= \frac{1}{(n!)^2} \sum_{\tilde{\sigma}_1 \in \mathbb{S}_n} \sum_{\tilde{\sigma}_2 \in \mathbb{S}_n} f_1(\sigma'\tilde{\sigma}_1) \otimes f_1(\sigma'\tilde{\sigma}_1\sigma) \otimes f_1(\sigma'\tilde{\sigma}_2) \otimes \big(\rho(\tilde{\sigma}_1)^\dagger \rho(\tilde{\sigma}_2)\big) \\
&= \frac{1}{(n!)^2} \sum_{\hat{\sigma}_1 \in \mathbb{S}_n} \sum_{\hat{\sigma}_2 \in \mathbb{S}_n} f_1(\hat{\sigma}_1) \otimes f_1(\hat{\sigma}_1\sigma) \otimes f_1(\hat{\sigma}_2) \otimes \big(\rho(\sigma'^{-1}\hat{\sigma}_1)^\dagger \rho(\sigma'^{-1}\hat{\sigma}_2)\big) \\
&= \frac{1}{(n!)^2} \sum_{\hat{\sigma}_1 \in \mathbb{S}_n} \sum_{\hat{\sigma}_2 \in \mathbb{S}_n} f_1(\hat{\sigma}_1) \otimes f_1(\hat{\sigma}_1\sigma) \otimes f_1(\hat{\sigma}_2) \otimes \big(\rho(\hat{\sigma}_1)^\dagger \rho(\sigma'^{-1})^\dagger \rho(\sigma'^{-1}) \rho(\hat{\sigma}_2)\big) \\
&= \frac{1}{(n!)^2} \sum_{\hat{\sigma}_1 \in \mathbb{S}_n} \sum_{\hat{\sigma}_2 \in \mathbb{S}_n} f_1(\hat{\sigma}_1) \otimes f_1(\hat{\sigma}_1\sigma) \otimes f_1(\hat{\sigma}_2) \otimes \big(\rho(\hat{\sigma}_1)^\dagger \rho(\hat{\sigma}_2)\big) \\
&= \mathcal{S}_{f_1}(\sigma, \rho)
\end{aligned}
$$

where we relabeled $\hat{\sigma}_1 = \sigma'\tilde{\sigma}_1$ and $\hat{\sigma}_2 = \sigma'\tilde{\sigma}_2$.  □

The key intuition here is that the functions of two isomorphic graphs are equal up to a translation by the permutation $\overline{\sigma}$ that links their respective adjacency matrices: let $f_\mathcal{G}$ be the function corresponding to the original graph and let $f_{\overline{\mathcal{G}}}$ be the function given by the permuted adjacency matrix, then $f_{\overline{\mathcal{G}}}(\sigma) = f_\mathcal{G}(\overline{\sigma}\sigma)$. Summing over all the elements of $\mathbb{S}_n$ neutralizes this translation.

## C   Extra experiments

### C.1   Synthetic dataset - weighted and directed

We repeated the same experiment of Section 3.1, with directed, weighted, and labeled graphs. Figure 2 reports the representatives of the four new families. Each edge has a weight in the interval $(0, 2]$.

Even in this case, the concatenation of the single orbit skew spectra allow the random forest to achieve an accuracy of $0.5$, while the 2-orbit skew spectra allow for an accuracy of $1$.

### C.2   Eigenvalue collisions

If we consider labeled graphs, then the eigenvalues or singular values, will be a valid invariant for them. Indeed a permutation matrix would only rotate the adjacency matrix, without changing the eigenvalues. However, it is easy to find examples of non-isomorphic labeled graphs where the eigenvalue invariant will collide. A valid example is the given by the graphs in Fig. 3.

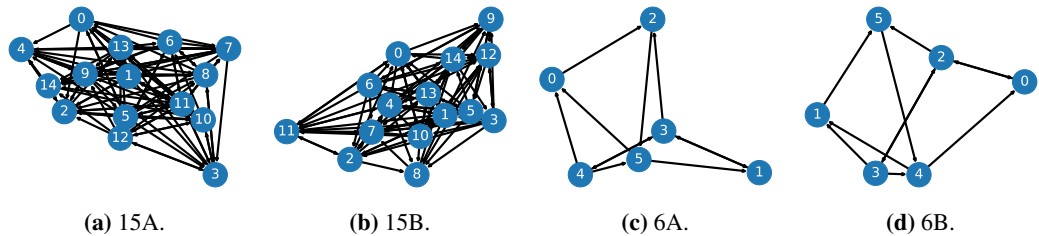

(a) 15A.      (b) 15B.      (c) 6A.      (d) 6B.

**Figure 2:** Synthetic dataset's graph families representatives. Graphs are directed and each edge has a weight in $(0, 2]$.

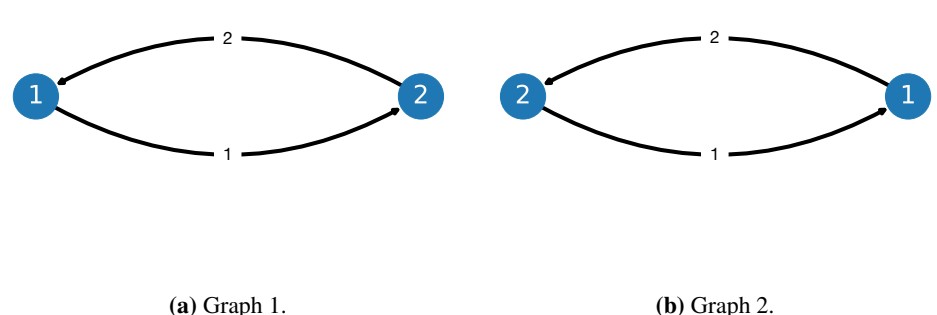

(a) Graph 1.          (b) Graph 2.

**Figure 3:** Example of two graphs with the same eigenvalues/singular values, but with distinct (reduced) skew spectra.

In the graph of Fig. 3a, we have that the node with label 1 goes to the one labeled 2 with a weight of 1. However, in the graph of Fig. 3b, the cost of moving from 1 to 2 is 2 and the two graphs are non-isomorphic. The two adjacency matrices of the graphs in Fig. 3 are

$$A_1 = \begin{bmatrix} 1 & 1 \\ 2 & 2 \end{bmatrix} \qquad\qquad A_2 = \begin{bmatrix} 2 & 1 \\ 2 & 1 \end{bmatrix}. \tag{8}$$

It is easy to verify that $\mathrm{Eigs}(A_1) = \{3, 0\}$ and $\mathrm{Eigs}(A_2) = \{3, 0\}$. Similarly, $\mathrm{SingVals}(A_1) = \mathrm{SingVals}(A_2) = \{3.16227766, 0\}$. However, the reduced skew spectra of the two graphs are different, meaning that the skew spectrum manages to distinguish the two cases.

