# OpenReview forum: "The Multi-Orbits Skew Spectrum: Boosting Permutation-Invariant Data Representations"
_logconference.io/LOG/2022/Conference — LoG 2022 Oral_

### Official Review · Reviewer_KxL4 · 2022-10-12

**Overall Score:** 6
**Confidence:** 4

**Review:**

**Extended Abstract Summary:**

The authors proposes a generalization of the skew spectrum of a graph [1] to multi-orbit cases and therefore handle cases of extracting features associated with labeled graphs, multi graphs and hypergraphs. Analogous to the single orbit case, the authors define an entry of the multi orbit skew spectrum using tensor products and leverage the strategy used in [1] to improve computational complexity.

**Strong Points:**
1. The generalization problem being tackled in this work is important, given that most real world graphs are weighted, labelled, have multiple relations or are hypergraphs.
2. Adapting [1] for the k=2 case (2 orbit case) in Section 2 is interesting. However, please see point 3 in weaknesses.
3. Experiment number 3.1 is well designed. Would recommend authors to add a similar experiment with directed graphs. Given the formulation of 3.2 as well, the adj matrix is weighted but symmetric - please show a synthetic use case, when this does not hold.


**Weak Points and Corresponding Questions and Suggestions:**
1. In my opinion, in its current form, the paper is written in a way which makes it artificially hard for the reader. For example, the authors define unlabeled graphs, but provide no concise definition of what is a labeled graph. On the other hand, authors provide an unnecessary definition of irreducible representation which is known well to audience with group/representation theory background. Would recommend the authors write a comprehensive background section in the appendix.
2. Line 60 - Claim of fairly easy to see needs to be substantiated. Do you mean just for undirected graphs? Otherwise it is not easy to see. Please add a simple proof in the appendix.
3. Line 73-76 -- Unfortunately, k=2 as a constant reduced the problem to 2-orbit cases and not the more general case as claimed in the title of the paper. Please provide Eqs (1-3) for the more general case and the associated computational complexity
3. Line 82-86 - the empirical evidence is not provided in the paper and the claim does not add much value to the reader without a more general proof or under what detailed limitations the empirical observations were made.
4. As pointed out by the authors, the generalization to hypergraphs with cardinalities of hyperedges > 3 are not scalable and hence impractical. Please add the computational complexity with multi relation (says k relations) graphs and for k-ary hypergraphs.


**Initial Recommendation:**

Weak Reject - the paper in current form is hard to read and the weak points above outweigh the strengths of the paper.

**References:**

[1]. Risi Kondor and Karsten M Borgwardt. The skew spectrum of graphs. In Proceedings of the 25th international conference on Machine learning, pages 496–503, 2008. 1, 2, 3

---

### Official Review · Reviewer_Axx2 · 2022-10-14

**Overall Score:** 8
**Confidence:** 4

**Review:**

This paper proposes and studies multi-orbit skew spectrum representation for incorporating additional symmetry information present in graphs that have such information. The contribution lies in generalizing/extending skew-spectrum computation to additional symmetry orbits.

I believe this paper should be accepted, as it presents a clear contribution that is relevant to the field. n particular, I believe that studies of spectral representations form an underexplored area in the field. The contribution is very clear, and the demonstrations are convincing.

Strong points:
1. The motivation and math are clear and consistent. The proposed approach is clean and easy to understand.
2. Synthetic experiment perfectly demonstrates main advantage of the method.
3. The paper is clearly written and easy to follow.

Weak points:
1. Scalability. As far as I understand, method is not scalable for graphs with many orbits (labels/node types/...) It seems to me (see Q. 1) that the computational complexity should depend on both k and n, unfortunately, the paper does not present it in the general setting.
2. The paper does not explain the relationship between the (multi-orbit) skew spectrum and its reduced counterpart. What is the information contained in the reduced skew spectra of graphs? How much less expressive are they?
3. There is no baseline, even the most simple one, for the QM7 experiment. I would suggest taking some very simple representation, such as the spectrum of the Adjacency or Laplacian matrices, just to give some perspective on the results.

Questions:
1. L73 - What is the computational complexity in for the general n, k?
2. What is the relationship between the reduced skew spectrum and its full variant?
3. Can reduced spectrum distinguish the graphs in the synthetic experiment? If so, it's probably a good idea to mention that.
4. Typo: Table 1: "Extreem" -> "Extreme"

---

### Official Review · Reviewer_qPAW · 2022-10-17

**Overall Score:** 6
**Confidence:** 3

**Review:**

Summary:

This paper considers generalizing the notion of a skew spectrum of a graph. The skew-spectrum is used for generating features of a graph which are invariant to permutations of the vertices and was introduced by Kondor and Borgwardt (2008) for machine learning tasks on graphs (such as classification and regression). The original skew spectrum only works for unlabeled graphs where only a single set of elements can permute amongst themselves (single orbit), while the generalization proposed herein works for unlabeled graphs, hypergraphs, multigraphs etc. where multiple sets of elements can permute within themselves (multiple orbits).

Pros: The generalization of the skew-spectrum to handle other types of graphs is an interesting result that to my knowledge seems novel and could be relevant for machine learning problems on graphs. The experiment conducted on the QM7 dataset seems to show a good improvement over the “single-orbit” skew spectrum approach, as shown in Table 1.

Cons:
(1)	The paper is unfortunately difficult to read as the exposition is not very clear. For a reader who is not familiar with the notion of a skew-spectrum, it is particularly difficult to understand even the very definition of a skew-spectrum in Section 1. This makes it difficult to properly understand the generalization proposed in Section 2. I had to go to the original paper of Kondor and Borgwardt (2008) in order to check the definitions. While the 4-page limit makes it difficult to go into details, it would have been nice to present some intuition as to how the skew-spectrum is defined.

(2)	 While the experiment in Section 3 seems to be promising, it is quite limited at the moment as only one dataset has been considered and no other competing methods have been compared. Therefore, it is difficult to ascertain the usefulness of the proposed approach.

Reasons for score:   The generalization of the skew-spectrum to handle more general families of graphs is a potentially useful contribution for the domain of machine learning on graphs. However, the paper lacks clarity in exposition and the experiments are a bit limited at the moment.

Other comments:

(1)	In Table 1, how are the error values calculated? This is not specified in the caption, or in Section 3.

(2)	In the experiment, what is the running time for the single orbit and 2-orbit representation-based approaches?

---

### Official Review · Reviewer_gs93 · 2022-10-26

**Overall Score:** 6
**Confidence:** 4

**Review:**

The paper investigates extensions of the skew spectrum to the cases of labeled graphs, multigraphs, and hypergraphs, which is an under-explored topic but very relevant given the complexity graphs can have.

This seems to be a solid theoretical contribution, but not necessarily by groundbreaking means: earlier results and theory are (potentially in a straightforward manner) extended for more complex graphs. The experimental evaluation shows that the multi-orbit skew spectrum outperforms the single-orbit.

I have a few concerns regarding "who is the audience" of this paper. I am not an expert in the skew spectrum and I had troubles following the paper. The notation and formulas are very dense and no intuition is given by what these formulas imply in practice. For example, following the math notation of the second paragraph in the introduction I would imagine that the skew spectrum is simply a permutation of the adjacency matrix. Of course there is nothing wrong with that, but at the moment the paper seems to target a very specific audience. With some further intuition, the paper will be accessible to a wider audience.

Further notes/comments:
- what does one lose by using the reduced spectrum?
- line 30: is it necessary for the subscript σ to be a function of n given that right afterwards the authors say that σ is in Sn?
- formula 1 is hard to parse (is this matrix concatenation?)
- line 98: shouldn't it be "same nodes"?
- what do the authors mean by their last sentence in the conclusion? How can they claim that their method is going to outperform the SOTA?

---

### Meta-Review · Area_Chair_Ms8x · 2022-11-16

**Confidence:** 4
**Recommendation:** Accept for spotlight

**Meta Review:**

The reviewers have assigned scores 6,6,6,8, so there is a consensus on accepting the paper. The method is based on a sophisticated understanding of representation theory / generalized Fourier theory, provides better results than 1-orbit features and other simple baselines, is relatively efficient to compute, and despite the sophisticated background should be easy to use for practitioners (since the features are invariant, one can apply any learning method on top of them).

The most significant limitations are:
1. Readability: reviewers gs93, qPAW, KxL4 note the difficulty to read, whereas Axx2 says the paper is easy to understand. It is clear that the paper assumes familiarity with representation theory, generalized Fourier transforms, and perhaps even prior exposure to the skew-spectrum of graphs. Given the page limit it will be challenging to introduce all of these in more detail, and at this point there are probably quite a few people in the community who have some familiarity with these topics. Given that all agree that the method is interesting and promising, it would be great to see a full paper with a more gentle exposition, but for the present 4-pager I find the paper entirely acceptable.
2. Experiments are limited: originally the paper did not compare to baselines. This has been remedied with some basic baselines. More experiments were promised for a followup paper.
3. Novelty: gs93 notes that "This seems to be a solid theoretical contribution, but not necessarily by groundbreaking means: earlier results and theory are (potentially in a straightforward manner) extended for more complex graphs". In my view the extension is straightforward/natural only to experts with a deep understanding of the skew-spectrum of graphs, and even for them working out all the details presented in this paper would be a significant amount of work.

Overall I think this paper contains a valuable contribution, with much promise for further work.

---

### Decision · Program_Chairs · 2022-11-23

Accept (Oral)